# Investigation of the Key Genes Associated with Anthocyanin Accumulation during Inner Leaf Reddening in Ornamental Kale (*Brassica oleracea* L. var. *acephala*)

**DOI:** 10.3390/ijms24032837

**Published:** 2023-02-02

**Authors:** Jiaqi Zou, Zhichao Gong, Zhiyong Liu, Jie Ren, Hui Feng

**Affiliations:** Liaoning Key Laboratory of Genetics and Breeding for Cruciferous Vegetable Crops, College of Horticulture, Shenyang Agricultural University, Shenyang 110065, China

**Keywords:** ornamental kale, leaf reddening, anthocyanin, metabolism, transcriptome

## Abstract

Ornamental kale (*Brassica oleracea* L. var. *acephala*) is a popular decorative plant in late autumn and winter. However, only during low-temperature color-changed periods below rough 15 °C can the plant accumulate anthocyanins and exhibit a diverse array of foliar color patterns. In this study, we probed into the potential mechanism of inner leaf reddening in a red-leaf pure line of ornamental kale by physiological, metabolic, and transcriptomic analyses. Determination of anthocyanin contents in the uncolored new white leaves (S0), the light red leaves (S1) in the reddening period and the red leaves (S2) completing color change, and analysis of anthocyanin metabolites at stage S2, revealed that the coloring of red leaves was mainly attributed to the accumulation of cyanidins. We further used transcriptomic sequencing between the pairwise S0, S1, and S2 stages to identify 21 differentially expressed genes (DEGs) involved in anthocyanin biosynthesis, among which the expression level of 14 DEGs was positively correlated with anthocyanin accumulation, and 6 DEGs were negatively correlated with anthocyanin accumulation. A total of 89 co-expressed genes were screened out, from which three DEGs (*BoCHI*, *Bo4CL3*, and *BoF3H*) were identified as hub genes in co-expression DEGs network. *BoDFR* and *BoCHI* were the DEGs with the highest expressions at S2. Moreover, two co-expressed DEGs related to stress response (*BoBBX17* and *BoCOR47*) also exhibited upregulated expressions and positive correlations with anthocyanin accumulation. A deep dive into the underlying regulatory network of anthocyanin accumulation comprising these six upregulated DEGs from S0 to S2 was performed via trend, correlation, and differentially co-expression analysis. This study uncovered the DEGs expression profiles associated with anthocyanin accumulation during ornamental kale inner leaf reddening, which provided a basis for further dissecting the molecular mechanisms of leaf color characteristic change in ornamental kale at low temperatures.

## 1. Introduction

Leaf color is of important commercial value for many ornamental plant species. Colored-leaf plants exhibit distinct color variations that result from the changes in pigment type and proportion. Among numerous leaf color variations, the formation of red leaves has received widespread attention, from which the vivid colorations could set off each other with the green part and embellish the leaves [1]. Plant red or pink leaves contain anthocyanin, chlorophyll, and carotenoid pigments. The reddening of leaves usually occurs after undergoing a period of low temperature, accompanied by decreases in chlorophyll and carotenoids contents and increases in anthocyanins [2].

Ornamental kale is a popular ornamental plant, with bright leaf colors, diverse patterns, strong cold resistance, and a long ornamental period [3]. Above all, ornamental kale requires low temperatures for developing its remarkable leaf color characteristic. Red-leaf ornamental kale is one of the most popular varieties, which is widely used to decorate in flower beds, flower borders, raised beds, and pots [4]. For ornamental kale, the synthesis and accumulation of anthocyanins principally cause the appearance of a red coloration when encountering low temperatures at the color turning stage during late autumn and winter [5,6].

Anthocyanins are water-soluble compounds that are naturally present in various plants with a wide range of biological functions. Anthocyanins furnish plant tissues and organs, such as petals, fruits, and seeds, with red, pink, purple, or blue colorations, and also render plants resistant to biotic and abiotic stresses [7,8]. In addition, anthocyanins possess strong antioxidant properties which are beneficial to human health [9]. The anthocyanin biosynthesis pathway has been intensively characterized, and most of the anthocyanin biosynthesis genes, including phenylalanine ammonia lyase (*PAL*), chalcone acetylase (*CHS*), chalcone isomerase (*CHI*) [10,11], flavanone 3-hydroxylase (*F3H*), flavonoid 3′-hydroxylase (*F3′H*), dihydroflavonol 4-reductase (*DFR*), anthocyanin synthase (*ANS*), and UDP-glucose: flavonoid 3-O-glucosyltransferase (*UFGT*), as well as transcription factor (TF) genes MYB, basic helix-loop-helix protein, WD40 protein, WRKY, and NAC TF, have been identified in higher plants [12,13].

Furthermore, the biosynthesis and accumulation anthocyanins are closely associated with environmental factors [14]. Temperature is a crucial external signal for regulating anthocyanin metabolism. In general, low temperatures can induce and activate the expression of genes related to anthocyanin synthesis, so as to lead to an increase in the content of anthocyanin in plants, while high temperatures could accelerate the degradation of anthocyanins, thereby leading to plant color fading. In addition, the accumulation of anthocyanins in some plants has been proved to be a strategy to adapt to low temperatures and other stresses [15,16,17,18,19].

Advances in comprehension of anthocyanin metabolism and molecular genetic mechanisms in colored leaves have been produced in recent years. More than 600 naturally occurring anthocyanins have been identified and isolated, most of which belong to six well-known anthocyanins: pelargonidin, cyanidin, delphinidin, peonidin, petunidin, and malvidin [20]. Red leaves mainly contain cyanidin and a small amount of malvidin, pelargonidin, and delphinidin [21,22,23,24,25]. Pink leaves mainly contain cyanidin-3-(erucic acid) (feruloyl)-diglucoside-5-glucoside [26]. It has been reported that purple leaves are dominant or incompletely dominant to pink or white leaves, red leaves are dominant or incompletely dominant to white or pink leaves, and the red leaf trait is controlled by genes with one or two alleles [27,28,29]. The trait for pink leaves is controlled by one incompletely dominant alleles and other alleles [30,31]. In ornamental kale, several genes that controlled some leaf colors have been mapped. We previously found dihydroflavonol reductase (*DFR*) on C09 controlled the red leaf trait [5,6]. Liu et al. (2017) and Feng et al. (2021) also found that DFR is a candidate gene for the control of purple and pink leaves in ornamental kale [32,33]. Yan et al. (2019) found that *BoMYB2* controls purple leaves in ornamental kale [34]. Jin et al. (2018) found that *BoC4H2*, *BoUGT9*, *BoGST21*, *BoHEMA*, *BoCRD1*, *BoPORC1*, *BoPORC2*, *BoCAO*, and *BoCLH1* play important roles in bicolor leaf formation in ornamental kale [35]. To date, the molecular mechanisms of ornamental kale red leaf formation in low-temperature environments still remain largely unclear. Guo et al. (2019) found that 10 anthocyanin biosynthesis genes (*BoDFR1*, *BoANS1*, *BoANS2*, *BoUGT79B1.1*, *BoTTG1*, *BoTT8*, *Bol012528*, *BoMYBL2.1*, *BoTT19.1*, and *BoTT19.2*) might participate in anthocyanin accumulation under low temperatures in ornamental kale [27]. However, the study only focused on the color change in new and mature leaves between different colored accessions, while the mechanism underlying anthocyanin accumulation in new leaves under low temperature remains unknown and warrants further exploration.

In this study, we detected pigment contents and anthocyanin metabolites in a red-leaf ornamental kale double haploid line ‘Y007-P-24′, in which the newly formed inner leaves undergo a coloration change from white to red at low-temperature color turning stage. To gain insights into the potential key genes involved in anthocyanin biosynthesis during the new leaves reddening under low temperature, a series of RNA sequencing and co-expressed gene analyses were performed among the three different leaf color development stages. Overall, the present study aimed to prospect key genes involved in anthocyanin accumulation and lay a foundation for further exploring molecular mechanisms underlying leaf color characteristics change in ornamental kale under low temperature.

## 2. Results

### 2.1. Anthocyanin Levels in the New Leaves of Red-Leaf Kale at Three Developmental Stages

During the leaf development of a red leaf ornamental kale ‘Y007-P-24′ (Figure 1a), its new leaves undergo a color change from white to red (Figure 1b). To investigate the new leaf reddening under low temperature, the anthocyanin contents of new leaves at three developmental stages (S0, S1, and S2) were detected using the pH differential method. As depicted in the Figure 1c, there were no anthocyanins detected at S0 and the total anthocyanin contents gradually increased from S0 to S2, which indicated that the new leaf reddening mainly resulted from the accumulation of anthocyanin.

### 2.2. Anthocyanin Metabolite Analysis

In order to further explore the mechanism of anthocyanin accumulation in the new red leaves, the component and content of anthocyanins at S2 were examined by LC-MS/MS. As showed in Table 1, a total of 23 anthocyanins were detected in red leaves, namely 6 cyanidins, 7 delphinidins, 1 malvidin, 2 pelargonidins, 5 peonidins, and 2 petunidins. Among them, the largest proportion was the cyanidins, accounting for 92.73% of the total content of anthocyanins, while the total content of malvidin only accounted for 0.02% of the total anthocyanin content. Above all, Cyanidin-3-O-glucoside is the highest component of all cyanidins, followed by Cyanidin-3-O-5-O-(6-O-coumaroyl)-diglucoside. These two abundant anthocyanin metabolites accounted for 77.30% of the total anthocyanins, suggesting that they might be the principal substances for the red coloration of ‘Y007-P-24′ new leaves.

### 2.3. Library Construction and DEG Analysis

For a better understanding of the molecular mechanisms underlying the color change in new leaves of ornamental kale, RNA-seq was exploited for comparing the differences between the pairwise developmental stages. Nine cDNA libraries (three biological replicates), corresponding to S0, S1, and S2, were constructed for transcriptomic sequencing. After filtering the raw reads, a total of 63.83 Gb valid clean data were obtained (Appendix A). There were 147,150,188, 140,048,122, and 138,348,338 valid reads identified in S0, S1, and S2, respectively (Appendix A). The clean reads yielded from all samples produced the bases scoring Q20 and Q30 values ranging from 99.36% to 99.73% and 95.34% to 96.89%, respectively. The GC content ranged from 46.5% to 47.00% (Appendix A). Approximately 87.12–88.92% of the reads were mapped to the reference genome. A total of 798, 3097, and 1333 differentially expressed genes (DEGs) were detected in S1 vs. S0, S2 vs. S0, and S2 vs. S1, respectively (Figure 2). Of the 798 DEGs in S1 vs. S0, 322 were upregulated, whereas 476 were downregulated. The 3097 DEGs in S2 vs. S0 contained 1615 upregulated DEGs and 1482 downregulated DEGs. A total of 673 upregulated and 660 downregulated DEGs were identified in S2 vs. S1. The DEGs in S1 vs. S0, S2 vs. S0, and S2 vs. S1 were used for trend analysis, gene expression pattern analysis, and co-expression analysis.

We subsequently analyzed the enriched Kyoto Encyclopedia of Genes and Genomes (KEGG) pathways of the DEGs (Figure 3). The DEGs in S1 vs. S0, S2 vs. S0, and S2 vs. S1 were mapped to 63, 112, and 88 pathways, respectively. Two pathways, including “flavonoid biosynthesis” (ko00941) and “phenylpropanoid biosynthesis” (ko00940), were identified in S1 vs. S0 and S2 vs. S0, which were closely related to anthocyanin accumulation in plant development. There were two, three, and four DEGs assigned to “flavonoid biosynthesis” (ko00941) with respect to S1 vs. S0, S2 vs. S0, and S2 vs. S1. *Bo8g081770* and *Bo9g177250* were upregulated expressed in all pairwise comparisons. Meanwhile, 5, 15, and 6 DEGs were mapped to “phenylpropanoid biosynthesis” (ko00940) in S1 vs. S0, S2 vs. S0, and S2 vs. S1, respectively. *Bo6g120830* was downregulated in all pairwise comparisons. Notably, the number of DEGs involved in these two pathways increased along with the accumulation of anthocyanin from S0 to S2.

### 2.4. Trend Analysis

All 3814 DEGs were filtered into eight distinct profiles. The DEGs in each profile involved a wide range of KEGG pathways and GO terms, and displayed similar expression patterns. Among these eight profiles, profiles 7, 0, 3, and 4 had a statistically significant number of DEGs assigned to them. Profile 7 contained the highest number of DEGs, followed by profile 0. The trend of DEGs in the profile 7 was positively correlated with anthocyanin accumulation, while negatively correlated with anthocyanin accumulation in the profile 0. Profile 7 contained 1029 DEGs involved in anthocyanin biosynthesis, chlorophyll biosynthesis, carotenoid metabolism, and photosynthesis. With regard to the 12 DEGs involved in anthocyanin biosynthesis, there were 10 structural genes (*BoC4H*, *Bo4CL3*, *BoF3H*, two *BoDFR*, *BoF3′H*, *BoCHI*, *BoCYP98A3*, *BoCCOAMT*, and *BoSAT*), 1 regulatory gene (*BoMYBL2*), and 1 transport gene (*BoTT19*). Profile 0 contained 877 DEGs, including 4 DEGs involved in anthocyanin biosynthesis (*BoSLP9*, *BoGL3*, *BoEGL3*, and *BoC4H*). Profile 4 also contained the gene involved in anthocyanin biosynthesis among its total 410 DEGs (Figure 4).

### 2.5. qRT-PCR to Determine Gene Expression Patterns

The gene expression patterns of the nine DEGs involved in anthocyanin metabolism were validated by qRT-PCR (Figure 5, Appendix A), which indicated that the transcriptome analysis results were reliable and could be used for further analysis.

### 2.6. Anthocyanin Biosynthesis Genes at S0–S2

We observed that the development process of new leaves gradually turning red attributed to the accumulation of anthocyanin in the leaves. On these grounds, we emphatically investigated the anthocyanin biosynthesis genes. In total, 95 genes involved in anthocyanin biosynthesis were singled out, including 48 structural genes, 44 regulatory genes, and 3 transport genes. The expression patterns of more than half of the genes involved in anthocyanin biosynthesis were similar to those involved in anthocyanin accumulation (Figure 6 and Figure 7). Of the 95 genes, 21 were identified as DEGs (Table 2). As for the 13 structural DEGs, 12 showed significantly higher levels at S2 or S1 than at S0, including *BoC4H* (*Bo5g052100*), *Bo4CL3* (*Bo6g099190*), *Bo4CL* (*Bo3g077430*), *BoCHS* (*Bo9g166290*), *BoCHI* (*Bo9g177250*), *BoF3H* (*Bo8g081770*), *BoF3H* (*Bo7g100840*), *BoF3′H* (*Bo9g174880*), *BoDFR* (*Bo9g058630* and *Bo2g116380*), *BoCCOAMT* (*Bo2g056370*), and *BoSAT* (*Bo3g042330*). With regard to the regulatory DEGs, five out of seven were downregulated at S1 or S2, including four positive regulatory genes and one negative regulatory gene. The four positive regulatory genes included *BoGL3* (*Bo4g141980* and *Bo4g141990*), *BoEGL3* (*Bo9g035460*), and *BoPR5* (*Bo6g119200*). The negative regulatory gene was *BoSLP9* (*Bo4g015800*). The upregulated DEGs included a positive regulatory gene, *BoHY5* (*Bo9g171430*) and a negative regulatory gene, *BoMYBL2* (*Bo2g070770*). In the aspect of the DEG involved in anthocyanin transport, there was only one that upregulated *BoTT19* (*Bo9g161480*) at S1.

The correlation coefficients (r) between the expression levels of DEGs and the contents of anthocyanin were calculated. The expression levels of 14 DEGs were positively correlated with anthocyanin accumulation, and the expression levels of 6 DEGs were negatively correlated with anthocyanin accumulation (Figure 6). The R^2^ values ranged from 0.0009 to 0.9980. Only the expression levels of two DEGs, *BoC4H* (*Bo5g052100*) and *Bo4CL3* (*Bo6g099190*), were significantly correlated with anthocyanin accumulation from S0 to S2 (R^2^ > 0.95, *p* ≤ 0.05). *BoDFR* and *BoCHI* were the two most highly expressed DEGs at S2.

Based on the two analyses above and the gene function assessment, *BoC4H* (*Bo5g052100*), *Bo4CL3* (*Bo6g099190*), *BoCHI* (*Bo9g177250*), *BoF3H* (*Bo8g081770*), *BoF3′H* (*Bo9g174880*), *BoDFR* (*Bo9g058630* and *Bo2g116380*), *BoCCOAMT* (*Bo2g056370*), *BoSAT* (*Bo3g042330*), *BoSLP9* (*Bo4g015800*), *BoHY5* (*Bo9g171430*), and *BoTT19* (*Bo9g161480*) might be responsible for anthocyanin accumulation during the development of new leaves from S0 to S2.

### 2.7. Identification of Differentially Co-Expressed Genes

To further identify the key genes involved in the new leaf reddening of ornamental kale, the DEGs in S1 vs. S0, S2 vs. S1, and S2 vs. S0 were compared, and 89 DEGs were identified as co-expressed genes (Figure 8a, Appendix A). These DEGs included *BoCOR47* (*Bo5g030290*) and *BoBBX17* (*Bo5g073010*), which were associated with stress response. In addition, three DEGs involved in anthocyanin biosynthesis were identified: *Bo4CL3*, *BoF3H,* and *BoCHI*. By comparing the upregulated and downregulated DEGs among three pairwise comparisons, it was found that there were 48 upregulated and 40 downregulated DEGs (Figure 8b,c). All of these 88 DEGs were identified as co-expressed genes in S1 vs. S0, S2 vs. S1, and S2 vs. S0. The DEGs, including *BoCOR47*, *BoBBX17*, *Bo4CL3*, *BoF3H,* and *BoCHI,* were all upregulated and co-expressed. Subsequently, we performed correlation analysis on the co-expressed DEGs and constructed a co-expression network, in which *Bo4CL3*, *BoF3H,* and *BoCHI* were identified as hub genes (Figure 8d; Appendix A). As mentioned above, the trends in the expression levels of these five DEGs were positively correlated with anthocyanin accumulation. Therefore, it was suggested that these genes might play a dominant role in anthocyanin accumulation of new leaves under low-temperature conditions.

## 3. Discussion

For ornamental kale, its newly formed inner leaves will gradually ‘blossom out’ into various pattern-colored leaves during the low-temperature color turning stage. This distinctive growth characteristic, together with its strong cold resistance, makes ornamental kale a popular decorative plant. Anthocyanins play an important role in plant physiology that furnish bright colors across the plant kingdom. Herein, we aimed to decipher the potential mechanism that underlies the new leaf reddening in ornamental kale. To this end, we firstly detected the changes of anthocyanin accumulation in new leaves at S0–S2 at the physiological level and determined the anthocyanin metabolites in detail. Following comparative transcriptome profiling between the pairwise S0, S1, S2 stages identified 21 DEGs involved in anthocyanin biosynthesis. A series of trend analysis, correlation analysis, and co-expression analysis were carried out. Four structural DEGs (*BoDFR*, *BoCHI*, *Bo4CL3*, and *BoF3H*) involved in anthocyanin biosynthesis and two transcription factor DEGs (*BoBBX17* and *BoCOR47*) in stress response were identified, which were upregulated when expressed from S0 to S2, and positively correlated with anthocyanin accumulation. Our results not only identified the key genes which might cooperatively contribute to the anthocyanins accumulation in ornamental kale at the color turning stage, but also provided insight into the potential mechanism underlying leaf reddening under low temperatures.

There are six types of anthocyanin pigments commonly found in nature, including pelargonidin, which usually renders plants orange-red in coloration; cyanidin, which gives plants a purplish-red coloration; and peonidin, which provides magenta color; while the blue-purple coloration is usually generated by delphinidin, petunidin, and malvidin [36]. The difference in these pigment proportions at the color-developing tissues ultimately determine the external coloration of plants. In the present study, the increased anthocyanin contents, along with the deepening of red coloration in new leaves, was almost 190 times higher at S2 than that at S0. Subsequent anthocyanin components analysis determined that the new leaf reddening was the result of the accumulation of cyanidins. High-level Cyanidin-3-O-glucoside and Cyanidin-3-O-5-O-(6-O-coumaroyl)-diglucoside were the two main components. Similar findings were highly consistent with that in the leaves of red cabbage (*B*. *oleracea*), which were predominantly constructed of contain cyanidin 3, 5-diglucoside [37,38]. Similarly, Cyanidin 3, 5-O-diglucoside played an important role in the final red coloration of *Acer pseudosieboldianum* in autumn [39]. These results suggest that the cyanidins play a dominant role in the red coloration of ornamental kale.

The anthocyanin biosynthesis pathway is the best-understood secondary metabolic pathway in plants and can be divided into four steps: phenylpropanoid biosynthesis, the polyketide pathway, early biosynthesis, and late biosynthesis. Anthocyanin accumulation is directly associated with structural and regulatory genes in the anthocyanin biosynthesis pathway [40]. Across plant species, anthocyanin biosynthesis was found to be regulated by MYB-bHLH-WD40 repeat protein (MBW) complexes [41]. To further explore the molecular mechanisms underlying anthocyanin biosynthesis and accumulation during the process of new leaves reddening in ornamental kale under low temperatures, we performed comprehensively comparative transcriptome analysis among different color development stages to prospect the key DEGs potentially making the greatest contributions to new leaf color variation. As a result, we identified 21 DEGs involved in anthocyanin biosynthesis from S0 to S2. The expression patterns of most structural genes were similar to those of genes involved in anthocyanin accumulation. Since the *Brassica* species experienced an extra whole genome triplication (WGT) event and tandem duplication (TD) compared with the model plant *Arabidopsis thaliana*, most of the anthocyanin biosynthetic genes were multiple gene copies in *B*. *oleracea* [27,42]. We noticed that several orthologous genes among these 21 DEGs exhibited different expression patterns. For example, *C4H* (*Bo5g052100*) transcript was increased from S0 to S2, while *C4H* (*Bo3g024650*) was the opposite. Another case was that two *DFR* (*Bo9g058630* and *Bo2g116380*) both showed up-regulated expressions in three comparison groups (Table 2). These results indicated that these duplicate anthocyanin biosynthesis genes might possess functional redundancy or play diverse roles in anthocyanin biosynthesis.

Of the differentially expressed 13 structural genes that were upregulated at S2 or S1 rather than at S0, three DEGs (*Bo4CL3*, *BoF3H*, and *BoCHI*) were denoted as co-expressed genes. Among them, we found that the expression pattern of *Bo4CL3* was significantly correlated with anthocyanin accumulation (Figure 6). 4CL is the key enzyme involved in the last step of the phenylpropanoid pathway and is expressed in many different organs. Up to now, four 4CL genes have been identified in *A. thaliana*; among these genes, *4CL3* is reportedly involved in flavonoid metabolism [43]. Guo et al. (2019) found *4CL3* exhibited a rather high expression level in purple inner leaves of a purple ornamental cabbage line [27]. In addition, in the transcriptome analysis of the roots from an anthocyanin-rich radish variety ‘Xinlimei’, *4CL3* together with the other nine structural genes were identified as critical for anthocyanin biosynthesis [44]. Notably, based on the conjoint analysis of transcriptomes and metabolomes in tolerant and sensitive rapeseed varieties, it was found that *Bn4CL3* as one of the strong candidate genes might function in adapting to cold stress in rapeseed [45]. Therefore, we conjectured that *Bo4CL3* might play a principal role in mediating anthocyanin accumulation at S0–S2 stages responding low temperatures.

*CHI* and *F3H*, as the ‘early’ genes in anthocyanin biosynthetic pathway, have been characterized in multiple species [46,47]. CHI can catalyze chalcone to naringenin, and the following F3H catalyzes the formation of colorless dihydroflavonols [48]. Studies have shown that a lack of *CHI* expression often results in decreased anthocyanin levels [49]. The expression levels of *CHI* were also different among the four flower development stages in the *Paeonia lactiflora* variety ‘Huangjinlun’ [50]. Forkman et al. (1980) reported a *f3h* mutant that led to the formation of white flowers in *Matthiolaincana* [51]. Furthermore, a knockout mutant of *F3H* in *A*. *thaliana* contained lower flavanol and anthocyanin contents compared to the wild type [52]. In addition, we noticed that the *BoDFR* accounting for the conversion of dihydrofavonols to leucoanthocyanidins in the late step of anthocyanin biosynthesis possessed the highest expression at S2. In ornamental kale, *DFR* has been proved as the causal gene conferring the purple, red, and pink leaf traits [5,6,32,33].

The accumulation of anthocyanins in multiple species has been proved as a strategy to adapt to various external stresses [40,53]. It has been reported that there might exist a strong relationship between low temperature induction and anthocyanin accumulation in plants, which can improve plant adaptation to low temperatures [19,54,55]. Therefore, we supposed that at low temperatures, the new leaves of ornamental kale might be induced to accumulate anthocyanin compounds. Previous studies of low-temperature-induced anthocyanin accumulation in Arabidopsis seedlings revealed that the *CHI*, *F3H*, and *DFR* were all up-regulated by low temperatures in a manner that was most partially or fully dependent on bZIP transcription factors *HY5*/*HYH*, which have been proved to be pivotal transcription factors in anthocyanin synthesis by regulating the promoter activity of the structural genes [56]. In our study, the positive regulatory gene *BoHY5* was also identified as an upregulated DEG in new leaf reddening, which might regulate the expression level of structural genes, including *BoCHI*, *BoF3H*, and *BoDFR*.

To further clarify the anthocyanin biosynthesis mechanism during the low-temperature color-changed period, we also identified the DEGs that respond to stress. Among the 89 co-expressed DEGs, *BoCOR47* and *BoBBX17* were identified as cold stress-response genes. The two DEGs were upregulated in all three comparisons. Trend analysis showed that these two DEGs, together with *Bo4CL3*, *BoCHI*, *BoF3H*, and *BoDFR*, all occurred in profile 7 and exhibited high correlation with these three hub DEGs (Figure 4a and Figure 8d; Appendix A). Guo et al. (1992) identified a cold-regulated wheat gene, *cor39*. This gene is related to *cor47* and is expressed in leaf, root, and crown tissues at low temperatures [57]. COR47 is one of the principal dehydrins (DHNs) proteins that accumulate in response to low temperatures and serve as a cryo-protector contributing to the cold stress response [58]. *COR47* overexpression and *RAB18* double-overexpression plants were cold tolerant [59]. In respect to *BBX17*, which belongs to B-box (BBX) zinc finger transcription factor family, it is involved in regulating the growth and development of plants and resisting various stresses [60,61]. Bai et al. (2014) found that an apple B-box protein controls the anthocyanin levels in peels under UV-B and low temperature conditions [62]. An et al. (2021) found that *MdBBX37* promoted apple cold tolerance by binding to the promoters of *MdCBF4* and *MdCBF4* and activating their transcription [63]. Three B-box proteins (BBX20, BBX21 and BBX22) served as essential partners for HY5-dependent modulation of anthocyanin accumulation and transcriptional regulation [64]. Gathering up these threads, we suspected that *BoCOR47*, *BoHY5*, *BoBBX17* might contribute to the anthocyanin accumulation, as well to improving cold tolerance. However, the exact role of *BBX17* and its potential linkage with *HY5* and *COR47* need to be further studied in ornamental kale.

Based on our results, we proposed a potential genetic mechanism of anthocyanin biosynthesis in the process of new leaf reddening of ornamental cabbage under low temperatures (Figure 9). It was speculated that low temperatures might induce the expression of *BoCOR47* and *BoBBX17* in the new white leaves at S0, whereafter the *BoBBX17* might further induce the up-regulated expression of *BoHY5*. The *BoHY5* could directly promote the synthesis of anthocyanins or indirectly promote the synthesis by activating the expression of MBW complex. The structural genes *Bo4CL3*, *BoF3H*, *BoCHI*, and *BoDFR* played a crucial role in anthocyanin accumulation during low-temperature new leaf reddening. The up-regulated *BoBBX17* and *BoCOR47* might act as cryo-protectors to adapt ornamental kale to cold environment. Overall, our study provided a comprehensive gene network of anthocyanin accumulation in ornamental kale, which contributes to further exploring molecular mechanisms underlying ornamental kale leaf color characteristics change under low temperatures.

## 4. Materials and Methods

### 4.1. Plant Materials

‘Y007-P-24′ is a red-leaf double haploid line from the Osaka Red variety (TAKII, Japan). The plants exhibited clear coloration with new red leaves during their decorative period when they were grown in a greenhouse at Shenyang Agricultural University (Shenyang, China). We sampled the leaves that were at three color developmental stages, which were harvested from the same plant at the same time (Figure 1b): S0 (white leaf), S1 (light red leaf with white main vein), and S2 (red and slightly green leaf with white main vein). New red leaves first appeared at the end of September to early October when the temperature fell below 15 °C. The phenotype appeared from the end of September to March of the following year. At S0–S2, samples of whole leaves without their main veins were collected, and three independent biological replicates were used. All samples were immediately frozen in liquid nitrogen and stored at −80 °C for pigment measurement, RNA-seq, and qRT-PCR.

### 4.2. Measurement of Anthocyanin Content

The total anthocyanin content was determined by measuring the absorbance at 536 and 700 nm in two buffers (pH 1.0 and 4.5) [65]. Data were obtained from three independent biological replicates. Total anthocyanin content was calculated using the following formulas:$$\mathrm{Anthocyanin}\;\mathrm{content}\;\left(\mathrm{mg}/\mathrm{mL}\right)=\frac{A\times MW\times DF}{e\times 1}$$

*A* = (A_536_ − A_700_) pH 1.0—(A_536_ − A_700_) pH 4.5, where *MW* = 449.2 g/mol; *e* = 26900; *DF* represents the dilution factor; 1 represents the optical path of the cuvette, which was 1 cm.

### 4.3. Anthocyanins Extraction and Multiple Reaction Monitoring

The freeze-dried samples were ground into powder (30 Hz, 1.5 min), and 50 mg powder was extracted with 0.5 mL methanol/water/hydrochloric acid (500:500:1, *v/v/v*). Then the samples were vortexed for 5 min, ultrasound for 5 min, and centrifuged at 12,000× *g* under 4 °C for 3 min. The residue was re-extracted by repeating the above steps under the same conditions. All the supernatants were collected and filtrated by a membrane filter (0.22 μm, Anpel) before LC-MS/MS analysis.

UPLC conditions: The sample extracts were analyzed using an UPLC-ESI-MS/MS system (UPLC, ExionLC™ AD, https://sciex.com.cn/, accessed on 1 March 2022); MS, Applied Biosystems 6500 Triple Quadrupole, https://sciex.com.cn/ (accessed on 1 March 2022). The analytical conditions were as follows: UPLC: column, WatersACQUITY BEH C18 (1.7 µm, 2.1 mm × 100 mm); solvent system, water (0.1% formic acid): methanol (0.1% formic acid); gradient program, 95:5 *v/v* at 0 min, 50:50 *v/v* at 6 min, 5:95 *v/v* at 12 min, hold for 2 min, 95:5 *v/v* at 14 min; hold for 2 min; flow rate, 0.35 mL/min; temperature, 40 °C; injection volume, 2 μL.

ESI-Q TRAP-MS/MS: Linear ion trap (LIT) and triple quadrupole (QQQ) scans were obtained on a triple quadrupole-linear ion trap mass spectrometer (QTRAP), QTRAP^®^ 6500+ LC-MS/MS System, equipped with an ESI Turbo Ion-Spray interface, operating in positive ion mode and controlled by Analyst 1.6.3 software (Sciex). The ESI source operation parameters were as follows: ion source, ESI+; source temperature 550 °C; ion spray voltage (IS) 5500 V; curtain gas (CUR) was set at 35 psi.

Anthocyanins contents were detected by Genepioneer based on the AB Sciex QTRAP 6500 LC-MS/MS platform, and were analyzed using scheduled multiple reaction monitoring (MRM). All the data were analyzed using Analyst 1.6.3 software (Sciex). All metabolites were quantified by Multiquant 3.0.3 software (Sciex). Mass spectrometer parameters, the declustering potentials (DP), and collision energies (CE) for individual MRM transitions were performed with further DP and CE optimization. A specific set of MRM transitions were monitored for each period according to the metabolites eluted within this period.

### 4.4. Identification of Anthocyanin Biosynthesis Genes in Brassica Oleracea

The sequences of anthocyanin biosynthesis genes in *A. thaliana* were downloaded from TAIR (http://www.arabidopsis.org/, accessed on 1 March 2022). The sequences were aligned with the relevant protein sequences using BLASTP. The threshold was set at an E-value ≤ 1 × 10^−10^.

### 4.5. RNA Extraction and Library Construction

Total RNA was extracted from each sample using a Total RNA Purification Kit (LC Science, Houston, TX, USA; TRK1001) according to the manufacturer’s protocol. Total RNA was extracted from each whole leaf without the main vein collected at S0, S1, and S2. Three biological replicates were used for each experiment. Nine RNA-seq libraries were constructed. All extracted RNA samples were further monitored for quantity and purity using a Bioanalyzer 2100 and an RNA 6000 Nano LabChip kit (Agilent Technologies, Santa Clara, CA, USA), with an RNA integrity > 7.0. Poly(A) mRNA was isolated from 10 μg RNA using the poly-T oligo method (Invitrogen, Carlsbad, CA, USA). Cleaved RNA fragments were used to generate a cDNA library using an mRNA-Seq Sample Preparation kit (Illumina, San Diego, CA, USA) following the manufacturer’s protocol. The 150-bp paired-end raw reads were sequenced by LC Science (Hangzhou, China) on an Illumina HiSeq 4000 platform. The adapters, low-quality reads, and ambiguous reads were removed from the raw reads. The Q20, Q30, and GC contents of the clean data were calculated (Appendix A).

### 4.6. Read Mapping onto the Reference Genome

All clean reads were mapped onto the *B. oleracea* reference genome (ftp://ftp.ensemblgenomes.org/pub/release-38/plants/genbank/brassica_oleracea/, accessed on 1 March 2022) using the HISAT package, allowing for a maximum of two mismatches and multiple alignments per read (up to 20 by default).

### 4.7. Differential Expression Analysis

All mapped reads were assembled using StringTie, and the fragments per kilobase million method was used to calculate mRNA expression levels. The DEGs were identified via log2 (fold change) > 1 and statistical significance (*p* ≤ 0.05). All DEGs were mapped to GO terms and KEGG pathways. Hypergeometric tests with the Bonferroni correction were used to find the significantly enriched KEGG pathways and GO terms. The threshold value was *p* ≤ 0.05.

### 4.8. Trend Analysis and Correlation Analysis

Trend analysis software was used to cluster the DEGs in S1 vs. S0, S2 vs. S0, and S2 vs. S1 into eight different expression profiles according to the log_2_Ratio (http://www.omicshare.com/tools/?l=en-us, accessed on 1 March 2022). Correlation analysis was performed using the OmicStudio tools at https://www.omicstudio.cn/tool/62 (accessed on 1 March 2022).

### 4.9. qRT-PCR

Total RNA was extracted from leaves at different developmental stages, as described above, and cDNA was synthesized from 2 μg total RNA using a cDNA synthesis kit (Vazyme, Nanjing, China) in a total volume of 20 μL. A total of nine DEGs related to anthocyanin biosynthesis in S0–S2 were selected for qRT-PCR. Gene-specific primers were designed using Primer Premier Software v.5.0 (Premier Biosoft, Palo Alto, CA, USA) (Appendix A). Ultra SYBR Mix (CWBIO, Beijing, China) and the QuantStudio 6 PCR system (Thermo Fisher Scientific, Waltham, MA, USA) was used for qRT-PCR. Furthermore, 50 μL reaction mixture contained 2 μL cDNA (1:50 dilution), 25 μL of 2× Ultra SYBR Mix (CWBIO, Beijing, China), and 1 µL of each primer (100 nM final concentration). The amplification conditions were as follows: 95 °C for 10 min, 40 cycles of 95 °C for 15 s, and 60 °C for 1 min. A melting curve analysis (55–95 °C) was performed at 95 °C for 15 s, 60 °C for 1 min, 95 °C for 15 s, and 60 °C for 15 s to confirm the specificity of the PCR amplification. Actin gene was used as an internal control. Relative expression levels were calculated using the 2^−ΔΔCt^ method [66]. All experiments were performed using three independent samples from three independent biological replicates.

### 4.10. Statistical Analysis

SPSS version 23 was applied to analyze the differences at different stages (IBM, New York, NY, USA). Analysis of variance (followed by Tukey’s test) was used to test differences between samples, with *p* ≤ 0.05 considered statistically significant. For correlation analysis, the Pearson correlation coefficient (r) was calculated.

## Figures and Tables

**Figure 1 ijms-24-02837-f001:**
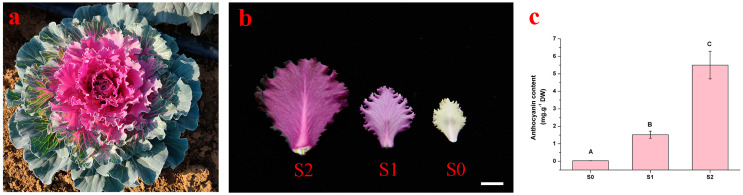
Phenotype and pigment accumulations in the new leaves at different developmental stages. (**a**) The whole plant morphology of ‘Y007-P-24′; (**b**) The new leaves phenotype of ‘Y007-P-24′ at different developmental stages. Bars = 1 cm; (**c**) The anthocyanin contents at S0–S2. Different capital letters show significant differences based on an analysis of variance (Tukey’s test, *p* ≤ 0.01).

**Figure 2 ijms-24-02837-f002:**
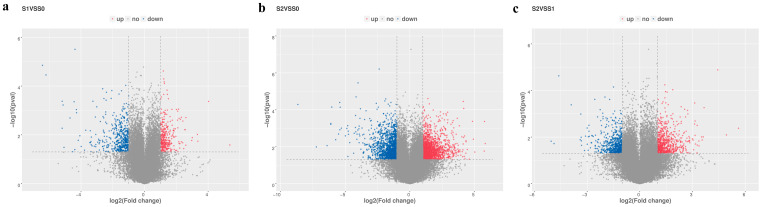
Differentially expressed genes in the pairwise comparisons. (**a**) DEGs in S1 vs. S0; (**b**) DEGs in S2 vs. S0; (**c**) DEGs in S2 vs. S1.

**Figure 3 ijms-24-02837-f003:**
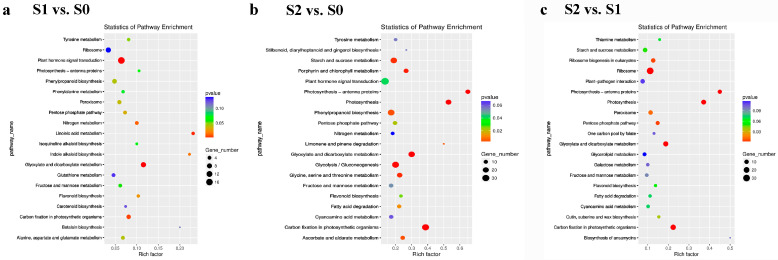
KEGG enrichment analysis for all three comparisons. (**a**) KEGG enrichment analysis for S1 vs. S0; (**b**) KEGG enrichment analysis for S2 vs. S0; (**c**) KEGG enrichment analysis for S2 vs. S1.

**Figure 4 ijms-24-02837-f004:**
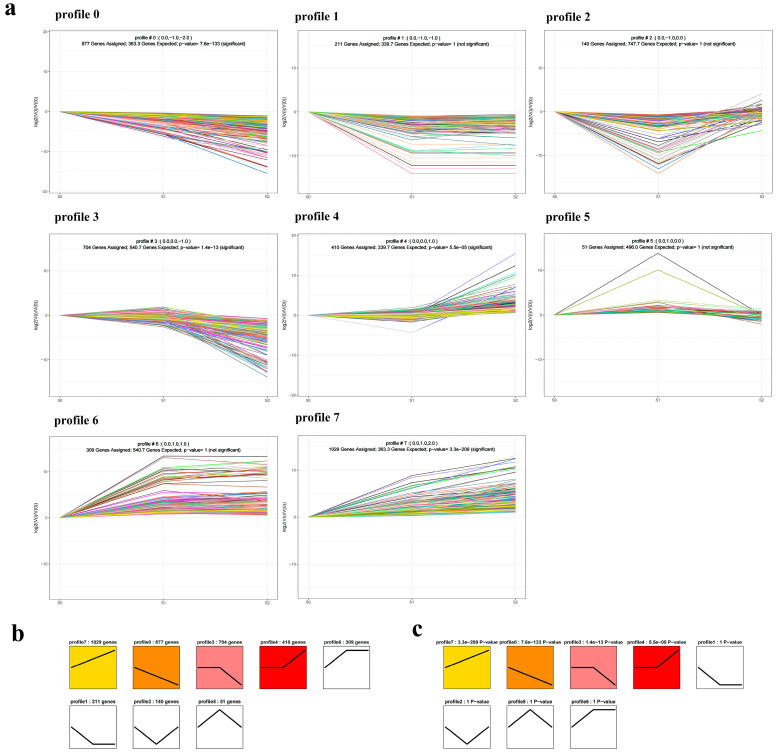
Trend analysis of differentially expressed genes (DEGs). (**a**) The trend analysis of all DEGs in all three pairwise comparisons. (**b**) The DEG numbers of eight distinct profiles; (**c**) The *p*-values of distinct profiles.

**Figure 5 ijms-24-02837-f005:**
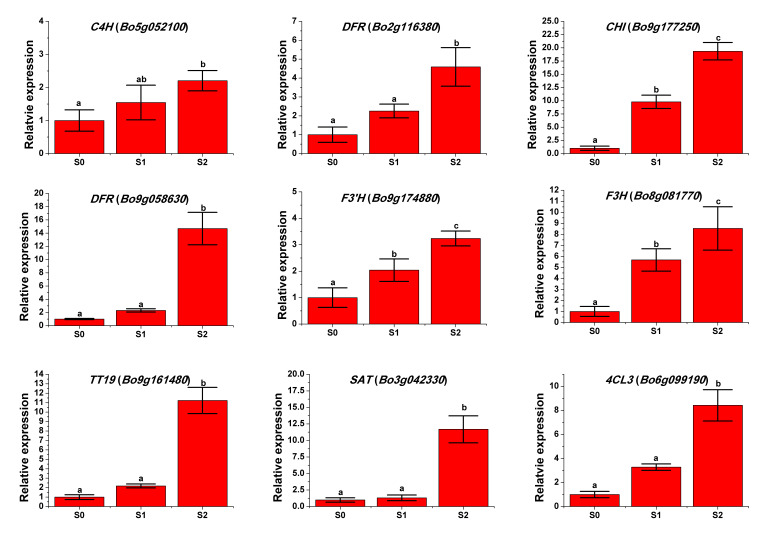
Expression levels of the DEGs related to anthocyanin biosynthesis at S0–S2. Data are represented as the mean ± standard deviation. Different lower-case letters indicate significant differences based on an analysis of variance (Tukey’s test, *p* ≤ 0.05).

**Figure 6 ijms-24-02837-f006:**
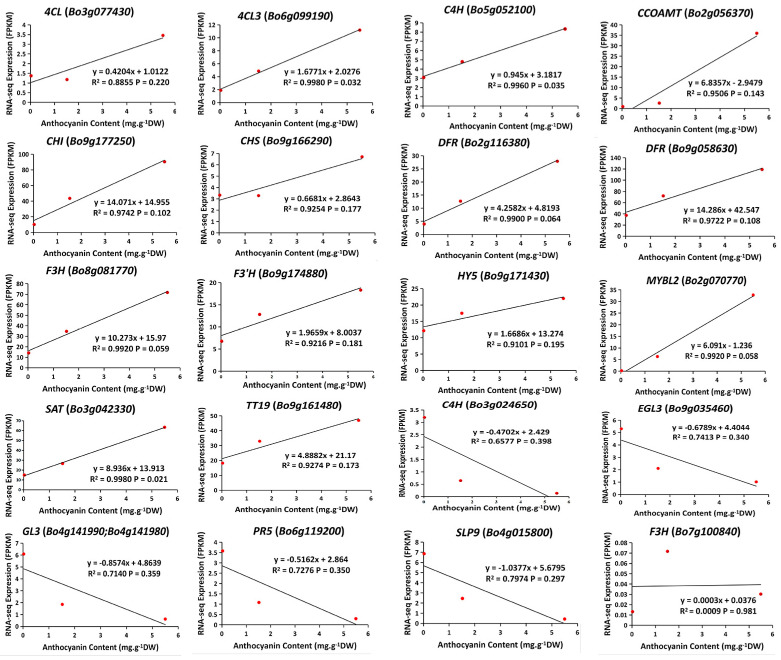
Correlation analyses between the relative expression levels of the genes and anthocyanin accumulation. Each panel represents a separate DEG.

**Figure 7 ijms-24-02837-f007:**
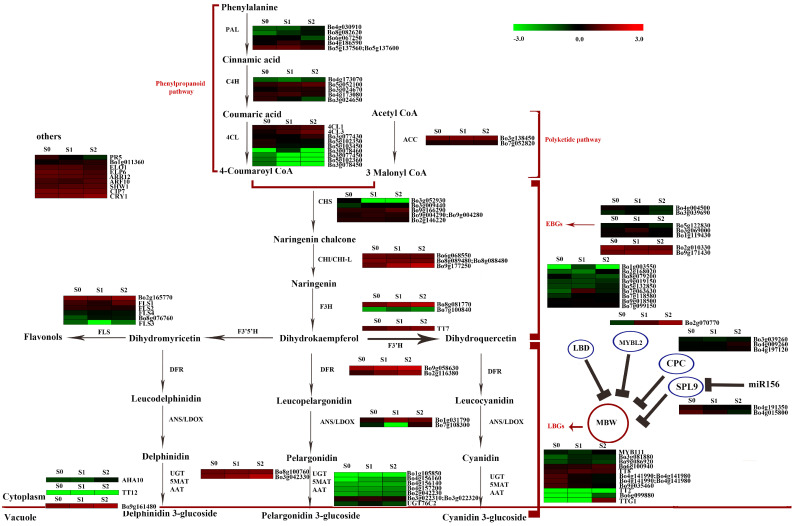
The expression levels of the genes involved in anthocyanin biosynthesis at each stage of leaf development (S0, S1, and S2). PAL: phenylalanine ammonia-lyase; C4H: cinnamic acid 4-hydroxylase; 4CL: 4-coumarate CoA ligase; CHS: chalcone synthase; CHI: chalcone isomerase; F3H: Flavanone 3-hydroxylase; F3′H: flavanoid 3′-hydroxylase; F3′5′H: flavonoid-3′,5′-hydroxylase; DFR: dihydroflavonol 4-reductase; ACC: acetyl CoA carboxylase; ANS: anthocyanidin synthase; FLS: flavonol synthesis; UGT: family 1 glycosyltransferases; 5MAT: anthocyanidin 5-O-glucoside-6″-O-malonyltransferase; AAT: anthocyanin acyltransferases; EBGs: early biosynthesis genes; LBGs: late biosynthesis genes; MBW: MYB-bHLH-WD40 complex; The expression levels of duplicated paralogs belonging to a protein family were clustered in the same heat map module.

**Figure 8 ijms-24-02837-f008:**
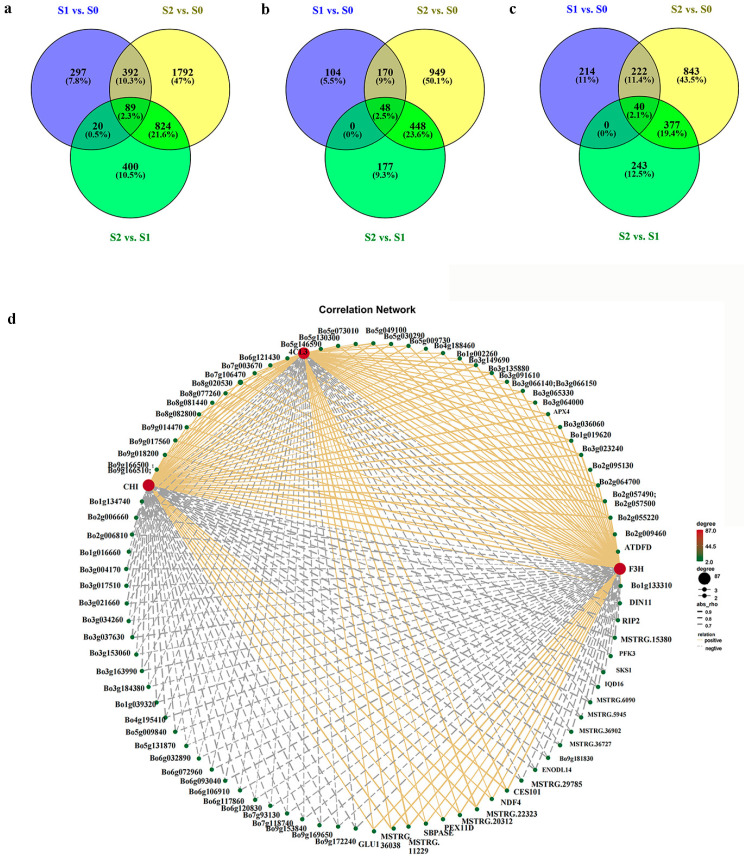
Co-expressed genes in the three development stages. (**a**) Co-expressed genes in the three comparisons; (**b**) Co-expressed upregulated DEGs in the three comparisons; (**c**) Co-expressed downregulated DEGs in the three comparisons; (**d**) The correlation network of co-expressed DEGs.

**Figure 9 ijms-24-02837-f009:**
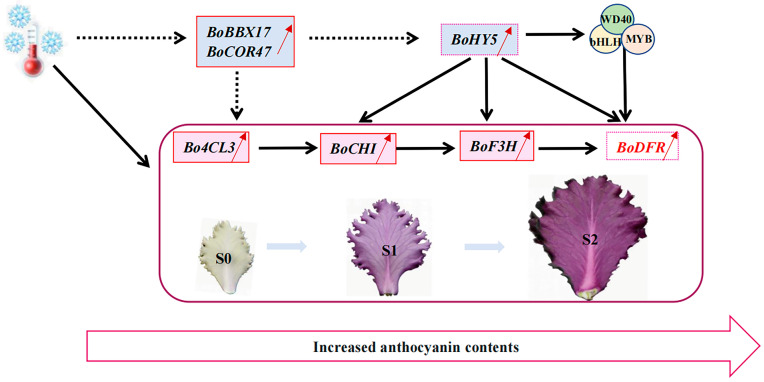
A possible regulatory network of anthocyanin accumulation in the process of new leaf reddening of ornamental kale under low temperature. Red solid box represents the upregulated DEGs in all three comparisons (S1 vs. S0, S2 vs. S1, and S2 vs. S0); Pink dashed box represents the upregulated DEGs in S2 vs. S0; Solid line indicates direct interaction; Dotted line indicates indirect interaction.

**Table 1 ijms-24-02837-t001:** Determination of the component and content of anthocyanins in the red leaves at S2.

Compounds	Class	Content (μg·g^−1^)
Cyanidin-3-(6″-caffeylsophoroside)-5-glucoside	Cyanidin	3.889 ± 0.304
Cyanidin-3-O-sambubioside-5-O-glucoside	Cyanidin	0.436 ± 0.077
Cyanidin-3-O-glucoside	Cyanidin	64.825 ± 4.807
Cyanidin-3-O-5-O-(6-O-coumaroyl)-diglucoside	Cyanidin	36.004 ± 2.653
Cyanidin-3,5,3-O-triglucoside	Cyanidin	5.515 ± 0.258
Cyanidin-3-O-sophoroside	Cyanidin	10.300 ± 0.954
Delphinidin-3-O-(6-O-malonyl-beta-D-glucoside)	Delphinidin	0.062 ± 0.006
Delphinidin-3-O-galactoside	Delphinidin	1.360 ± 0.208
Delphinidin-3-O-(6-O-acetyl)-glucoside	Delphinidin	0.017 ± 0.005
Delphinidin-3-O-sophoroside	Delphinidin	0.964 ± 0.073
Delphinidin-3-O-sambubioside	Delphinidin	0.015 ± 0.0005
Delphinidin-3-O-rhamnoside	Delphinidin	0.138 ± 0.142
Delphinidin-3,5-O-diglucoside	Delphinidin	2.365 ± 0.223
Malvidin-3-O-glucoside	Malvidin	0.030 ± 0.003
Pelargonidin-3-(6″-caffeylsophoroside)-5-glucoside	Pelargonidin	0.008 ± 0.002
Pelargonidin-3-sophoroside-5-glucoside	Pelargonidin	1.577 ± 0.258
Peonidin-3-(caffeoyl-glucosyl-glucoside)-5-glucoside	Peonidin	0.010 ± 0.003
Peonidin-3-O-(6″-ferulylsophoroside)-5-glucoside	Peonidin	0.003 ± 0.001
Peonidin-3-O-glucoside	Peonidin	0.131 ± 0.019
Peonidin-3,5-O-diglucoside	Peonidin	0.536 ± 0.042
Peonidin-3-sophoroside-5-glucoside	Peonidin	1.105 ± 0.224
Petunidin-3-O-sophoroside	Petunidin	0.877 ± 0.133
Petunidin-3-O-(6-O-malonyl-beta-D-glucoside)	Petunidin	0.277 ± 0.061

Data are represented as the mean ± standard deviation. (*n* = 3).

**Table 2 ijms-24-02837-t002:** Differently expressed genes related to anthocyanin biosynthesis at S1 vs. S0, S2 vs. S1 and S2 vs. S0.

Gene	Annotation	S1 vs. S0	S2 vs. S1	S2 vs. S0
*Bo5g052100*	*C4H*	up	up	up *
*Bo3g024650*	*C4H*	down	down	down *
*Bo6g099190*	*4CL3*	up *	up *	up *
*Bo3g077430*	*4CL*	up	up *	up *
*Bo9g166290*	*CHS*	up	up *	up
*Bo9g177250*	*CHI*	up *	up *	up *
*Bo8g081770*	*F3H*	up *	up *	up *
*Bo7g100840*	*F3H*	up	down	up *
*Bo9g174880*	*F3′H*	-	up	up *
*Bo9g058630*	*DFR*	up	up	up *
*Bo2g116380*	*DFR*	up *	up	up *
*Bo3g042330*	*SAT*	up *	up	up *
*Bo4g141980*	*GL3*	down	down	down *
*Bo4g141990*	*GL3*	down	down	down *
*Bo9g035460*	*EGL3*	down	down	down *
*Bo9g171430*	*HY5*	up	up	up *
*Bo2g070770*	*MYBL2*	up *	up	up *
*Bo4g015800*	*SLP9*	down	down	down *
*Bo9g161480*	*TT19*	up *	up	up
*Bo6g119200*	*PR5*	down	down *	down *
*Bo2g056370*	*CCOAMT*	up	up *	up *

* Represents differentially expressed genes.

## Data Availability

The datasets generated for this study can be found in the NCBI sequence reads archive (SRA) database under BioProject No. PRJNA898659.

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
