# Peer review of "Investigation of the Key Genes Associated with Anthocyanin Accumulation during Inner Leaf Reddening in Ornamental Kale (Brassica oleracea L. var. acephala)"

_ijms, 2023, doi:10.3390/ijms24032837_

Round 1

Reviewer 1 Report

This study is well conducted with clear objectives. Informative findings are well presented and discussed. I have a few questions and some minor comments which could be clarified or addressed for improving the general readership of this manuscript.

Line 133 More description about deep transcriptomic sequencing should be added in the methods section 4.5. In other words, why is it called deep transcriptomic sequencing? What’s the difference from the regular transcriptomic sequencing?

Table 2. Some DEGs are annotated as having the same function. How did the authors interpret that since the study material is a double haploid line? For those genes with the same annotation, for example, both Bo09g058630 and Bo2g116380 are DFR. Which one is given priority for future studies?

Are there even more genes annotated as the same enzyme in the anthocyanin pathway (Figure 7)? Please clarify or briefly describe them in figure legend.

Figure 6 What’s the cutoff value for R2 significance? This question is based on the statement “Only the expression levels of BoC4H (Bo5g052100) and Bo4CL3 (Bo6g099190) were significantly correlated with anthocyanin accumulation.” in line 219-220.

Line 263 Change ‘4 DEGs’ to ‘Four structural DEGs’.

Line 340 Is there a supplementary file for all these 89 co-expressed DEGs?

Figure 9 What do the dotted line, solid line indicate? directly or indirectly?  Also, what’s the meaning of dotted box and solid box for those genes? Please clarify.

Section 4.7 and 4.8 can be combined.

Author Response

28th Jan, 2023

Dr. Sureerat Namken

Editor

Dear Editor:

I wish to re-submit the manuscript titled “Investigation of the key genes associated with anthocyanin accumulation during inner leaf reddening in ornamental kale (Brassica oleracea var. acephala).” The manuscript ID is ijms-2188102.

The manuscript has been rechecked carefully and the changes have been made in accordance with the reviewers’ suggestions. The responses to all comments have been summarized below. All revisions in the manuscript are be marked up for easy identification.

We thank you and the reviewers for your thoughtful suggestions and insights. The manuscript has benefited from your suggestions. I look forward to working with you to move this manuscript closer to publication in International Journal of Molecular Sciences.

Sincerely,

Hui Feng

College of Horticulture,

Shenyang Agricultural University

120 Dongling Road Shenhe District Shenyang P.R. China.

Email: fenghuiaaa@syau.edu.cn

To Reviewer 1:

Comments to the Author:

This study is well conducted with clear objectives. Informative findings are well presented and discussed. I have a few questions and some minor comments which could be clarified or addressed for improving the general readership of this manuscript.

  • Line 133 More description about deep transcriptomic sequencing should be added in the methods section 4.5. In other words, why is it called deep transcriptomic sequencing? What’s the difference from the regular transcriptomic sequencing?

Response: We thank the Reviewer for bringing this to our attention. Your insightful comments made us perceive that the ‘deep’ might be ambiguous for describing transcriptomic sequencing. To be more rigorous, we have deleted the word and rechecked the descriptions about transcriptomic sequencing in the manuscript. In our study, nine cDNA libraries (three biological replicates) corresponding to S0, S1 and S2 were constructed for transcriptomic sequencing. After filtering the raw reads, a total of 63.83-Gb valid clean data were obtained. The Q20 (>99.36%), Q30 (>95.34%), and GC contents (46.5%-47%) of the data were calculated. Our comparative transcriptome analyses were based on these high-quality clean data. As you suggested, the Results section 2.3 the Methods section 4.5 has been refined. (Lines 133-142; 479-480). The summary of transcriptome sequencing data mentioned above has been supplemented as a new Table S1 as below.

Table S1 Summary of transcriptome sequencing data

Sample

Raw Data Read

Raw Data Base

Valid Data Read

Valid Data Base

Valid Ratio (reads)

Q20%

Q30%

GC content%

S0_1

45,944,336

6.89G

45,381,766

6.81G

98.78

99.68

96.89

46.5

S0_2

46,147,772

6.92G

45,672,436

6.85G

98.97

99.63

96.79

46.5

S0_3

56,896,248

8.53G

56,095,986

8.41G

98.59

99.5

95.34

46.5

S1_1

46,366,224

6.95G

44,788,004

6.72G

96.6

99.36

95.82

46.5

S1_2

46,445,136

6.97G

45,910,946

6.89G

98.85

99.52

96.45

47

S1_3

52,275,366

7.84G

49,349,172

7.40G

94.4

99.38

96.07

46.5

S2_1

49,100,766

7.37G

48,519,316

7.28G

98.82

99.54

96.43

46.5

S2_2

41,965,700

6.29G

41,491,120

6.22G

98.87

99.48

96.44

47

S2_3

48,852,286

7.33G

48,337,902

7.25G

98.95

99.73

95.36

47

  • Table 2. Some DEGs are annotated as having the same function. How did the authors interpret that since the study material is a double haploid line? For those genes with the same annotation, for example, both Bo09g058630and Bo2g116380 are DFR. Which one is given priority for future studies?

Response: The ‘Y007-P-24’ was screened from the double haploid plant population of Osaka red variety, and its newly white inner leaves exhibited distinct red deepening without interference from other variegation, so it was used as experimental material in this study. The pure line ‘Y007-P-24’ is beneficial to the preservation of leaf-color trait. Since the Brassica species experienced an extra whole genome triplication (WGT) event and tandem duplication (TD) compared with the model plant Arabidopsis thaliana, there would be three or more copies of genome blocks in B. oleracea when only one copy of relative genome block in Arabidopsis. Therefore, some DEGs in ornamental kale are annotated as having the same function in anthocyanins biosynthesis.

For the two duplicated paralogs (Bo09g058630 and Bo2g116380) in B. oleracea, our previous studies of fine-mapping the Re controlling red leaf trait in ornamental kale have identified Bo09g058630 as the candidate gene; Besides, Bo09g058630 was also shown to confer the purple leaf trait in other cultivars of ornamental kale. It recently was identified as giving pink-leaved ornamental kale its color. However, the role of another paralog (Bo2g116380) remains largely unknown. Therefore, we would give priority to deciphering the function of Bo2g116380 in our future studies. To be clearer and in accordance with the Reviewers’ concern, we have added the above relevant content to Discussion section (Lines 320-329).

  • Are there even more genes annotated as the same enzyme in the anthocyanin pathway (Figure 7)? Please clarify or briefly describe them in figure legend.

Response: We agree with the Reviewer’s comments that there are several genes that are annotated as encoding the same category enzyme in the anthocyanin pathway, which could be arise by whole genome triplication (WGT) event or tandem duplication (TD). In Figure 7, the expression levels of duplicated paralogs belonging to a protein family were clustered in the same heat map module. As you suggested, we have clarified this in figure legend of Figure 7 (Lines 244-251).

  • Figure 6 What’s the cutoff value for R2significance? This question is based on the statement “Only the expression levels of BoC4H (Bo5g052100) and Bo4CL3 (Bo6g099190) were significantly correlated with anthocyanin accumulation.” in line 219-220.

Response: When the value of R2 is greater than 0.95 and at significant difference level p<0.05, there would be a significantly positive correlation between the relative expression levels of DEGs and anthocyanin accumulation. We have supplemented this in revised manuscript in the light of your suggestions (Lines 233-235).

  • Line 263 Change ‘4 DEGs’ to ‘Four structural DEGs’.

Response: We have changed ‘4 DEGs’ to ‘Four structural DEGs’ (Line 286).

  • Line 340 Is there a supplementary file for all these 89 co-expressed DEGs?

Response: We have added the information of all 89 co-expressed DEGs as the Supplementary file 1.

  • Figure 9 What do the dotted line, solid line indicate? directly or indirectly?  Also, what’s the meaning of dotted box and solid box for those genes? Please clarify.

Response: We agree with the Reviewer’s comments. Red solid box represents the upregulated DEGs in all three comparisons (S1 vs. S0, S2 vs. S1 and S2 vs. S0); Pink dashed box represents the upregulated DEGs in S2 vs. S0; Solid line indicates direct interaction; Dotted line indicates indirect interaction. We have added these annotations in the legend of Figure 9 (Lines 409-412).

  • Section 4.7 and 4.8 can be combined.

Response: We have combined these two parts into a new section 4.7 according to your suggestions (Lines 488-491).

Reviewer 2 Report

The aim of this study is to establish Brasica oleraea L. var. acephala genes  responsible for leaf reddening after induction with low temperature. Study is very technical, narrow aimed, but has some element of novelty.

I have some remarks:

quality of Fig. 4 is very low, high-resolution picture must be submitted.

Table 1 lacks description of statistics

In my opinion correlational analysis with 3 data points doesn’t allow to make significant conclusions, it may show some trend, but not direct relation between studied parameters.

Fig. 7 should be MBW complex

I doubt, that accumulation of anthocyanins may allow for plant to adapt to cold, reduced temperature activates not only anthocyanin biosynthesis pathway, but also a number of other pathways related to cold acclimation (i.e. DREB1/CBF). You see strong correlation, because low temperature initiates anthocyanin biosynthesis, but that doesn’t mean anthocyanins act as a cryoprotectors. Cor47 codes dehydrins, which serve as cryoprotector in plants, and they are activated by low temperature as well. I suggest removing speculations on anthocyanin role as enhancers of plant adaptivity to cold.

In general, I think manuscript may be published after minor revision.

Author Response

28th Jan, 2023

Dr. Sureerat Namken

Editor

Dear Editor:

I wish to re-submit the manuscript titled “Investigation of the key genes associated with anthocyanin accumulation during inner leaf reddening in ornamental kale (Brassica oleracea var. acephala).” The manuscript ID is ijms-2188102.

The manuscript has been rechecked carefully and the changes have been made in accordance with the reviewers’ suggestions. The responses to all comments have been summarized below. All revisions in the manuscript are be marked up for easy identification.

We thank you and the reviewers for your thoughtful suggestions and insights. The manuscript has benefited from your suggestions. I look forward to working with you to move this manuscript closer to publication in International Journal of Molecular Sciences.

Sincerely,

Hui Feng

College of Horticulture,

Shenyang Agricultural University

120 Dongling Road Shenhe District Shenyang P.R. China.

Email: fenghuiaaa@syau.edu.cn

To Reviewer 2:

Comments to the Author:

The aim of this study is to establish Brassica oleracea L. var. acephala genes responsible for leaf reddening after induction with low temperature. Study is very technical, narrow aimed, but has some element of novelty. I have some remarks:

  • Quality of Fig. 4 is very low, high-resolution picture must be submitted.

Response: We thank the Reviewer for pointing this out. We have resubmitted a high-resolution Fig. 4 in the revised manuscript (Line 187).

  • Table 1 lacks description of statistics.

Response: We have added the description of statistics for Table 1 (Line 131). The compound content of each anthocyanin metabolite was represented by Mean ± SD (n=3).

  • In my opinion correlational analysis with 3 data points doesn’t allow to make significant conclusions, it may show some trend, but not direct relation between studied parameters.

Response: We agree with the Reviewer’s comments that the correlational analysis could reflect the proximity degree of expression trend between hub gene and other data points. In line with your suggestions, the conclusions about significance have been deleted (Line 266).

  • 7 should be MBW complex.

Response: We have revised this label in Fig.7 (Line 241) as you suggested.

  • I doubt, that accumulation of anthocyanins may allow for plant to adapt to cold, reduced temperature activates not only anthocyanin biosynthesis pathway, but also a number of other pathways related to cold acclimation (i.e. DREB1/CBF). You see strong correlation, because low temperature initiates anthocyanin biosynthesis, but that doesn’t mean anthocyanins act as a cryoprotectors. Cor47 codes dehydrins, which serve as cryoprotector in plants, and they are activated by low temperature as well. I suggest removing speculations on anthocyanin role as enhancers of plant adaptivity to cold. In general, I think manuscript may be published after minor revision.

Response: We thank the Reviewer for raising this concern. We agree with the Reviewer’s comments that low temperature initiates anthocyanin biosynthesis in ornamental kale for its typically leaf color changing characteristics at low temperatures and the differential expression of key genes related to anthocyanin accumulation identified in our study. In line with your insightful suggestions, it is most likely that ornamental kale adaptation to cold might benefit from the function of cold acclimation-related genes. We have removed the speculation that anthocyanins accumulation might enhance cold tolerance in ornamental kale (Lines 360-361) and have supplemented the function of Cor47 in plants cold response (Lines 378-380), and the explanation for possible regulatory network has been revised (Lines 400-403).
